# Just Keep Rolling?—An Encompassing Review towards Accelerated Vaccine Product Life Cycles

**DOI:** 10.3390/vaccines11081287

**Published:** 2023-07-27

**Authors:** Janis Stiefel, Jan Zimmer, Jeffrey L. Schloßhauer, Agnes Vosen, Sarah Kilz, Sascha Balakin

**Affiliations:** 1Fraunhofer Institute for Microengineering and Microsystems IMM, Carl-Zeiss-Straße 18-20, 55129 Mainz, Germany; 2Fraunhofer Institute for Cell Therapy and Immunology, Branch Bioanalytics and Bioprocesses IZI-BB, Am Mühlenberg 13, 14476 Potsdam, Germany; 3Fraunhofer Center for International Management and Knowledge Economy IMW, Neumarkt 20, 04109 Leipzig, Germany; 4Fraunhofer Institute for Ceramic Technologies and Systems IKTS Material Diagnostics, Bio- and Nanotechnology, Maria-Reiche-Straße 2, 01109 Dresden, Germany; 5Max Bergmann Center of Biomaterials (MBC), Technical University of Dresden, Budapester Strasse 27, 01069 Dresden, Germany

**Keywords:** vaccine adaption, product life cycle, nanocarrier, mRNA vaccines, vaccine market, protein structure prediction, digital twin

## Abstract

In light of the recent pandemic, several COVID-19 vaccines were developed, tested and approved in a very short time, a process that otherwise takes many years. Above all, these efforts have also unmistakably revealed the capacity limits and potential for improvement in vaccine production. This review aims to emphasize recent approaches for the targeted rapid adaptation and production of vaccines from an interdisciplinary, multifaceted perspective. Using research from the literature, stakeholder analysis and a value proposition canvas, we reviewed technological innovations on the pharmacological level, formulation, validation and resilient vaccine production to supply bottlenecks and logistic networks. We identified four main drivers to accelerate the vaccine product life cycle: computerized candidate screening, modular production, digitized quality management and a resilient business model with corresponding transparent supply chains. In summary, the results presented here can serve as a guide and implementation tool for flexible, scalable vaccine production to swiftly respond to pandemic situations in the future.

## 1. Introduction

Conventional development and manufacturing of vaccines against infectious diseases is complex and involves many suppliers, taking up to 15 years from development to the approval of a vaccine candidate [1]. Younger companies, such as Moderna (2010) and BioNTech (2008), have also been addressing vaccination in the field of cancer for years (BioNTech iNest BNT122; Moderna mRNA-4157/V940). Thus, they are setting a clear course toward personalized medicine, which has a significantly shorter product life cycle and involves greater product variability than previous vaccines and drugs. In view of the increasing competition due to the “New Era of Medicine” (mRNA drugs made popular by the pandemic), rapid adaptation of already existing vaccines is becoming increasingly important and driving the market situation to a new level.

In 2019, the global vaccine market made up approximately 2.5% of the total global pharmaceutical industry [2], making it one of the top ten therapeutic areas in terms of revenue [3]. At that time, the market comprised 5.5 billion vaccine doses, with sales of $33 billion [2]. About three-quarters of global vaccine production took place and still takes place in Europe, with the majority of vaccines produced being exported [4]. The largest customer regions are Southeast Asia, Africa and the USA [2].

In addition, the global vaccine market is characterized by an oligopolistic structure [5]. Since 1996, numerous mergers or acquisitions of pharmaceutical companies have led to a strong concentration in the vaccine market [6]. Therefore, with GlaxoSmithKline, Merck & Co, Sanofi and Pfizer, four key companies now take more than 91% of the global market. The high costs, time and uncertainty of regulatory conditions related to the production of vaccines limit small- and medium-sized companies, as they do not have the necessary skills and resources. However, since the emergence of the COVID-19 pandemic, the industry has experienced a boost in innovation: research and development investments, as well as government participation in the market, have significantly increased [7], which has also brought new players with innovative solutions to the market [8]. The APAC region in particular has been able to strengthen its position in the market [5] by rapidly adapting its vaccine development and production as well as its supply chain network to the new reality [7]. Sales of traditional vaccines are expected to continue to grow over the next five years [9], but market concentration will decline due to an increased number of competitors [5]. The increase in various infectious diseases [10], as well as investments in the research and development of vaccines, will be a driver for growth. In addition, investments in new therapeutic areas and improved production methods are expected to increase [7]. The demand for low-cost vaccines adapted to the needs of developing countries represents a major opportunity for vaccine developers [5]. Already, it is evident that vaccine manufacturers are pursuing diversified strategies to expand their geographic footprint and optimize their operations for more efficient and profitable vaccine production [7].

The quantum leaps in technology that accompanied the rapid market development have, not least, revealed great potential for improvement in accelerating the vaccine product life cycle. This review is intended to provide a holistic view of the vaccine adaptation process from the pharmacological level to production and logistical and economic issues with consideration of economic impacts and framework conditions. Therefore, the bottlenecks of the conventional vaccine production chain were identified and aligned with current innovation approaches based on research in the literature, ecosystem analysis and qualitative interviews with key players to bring the broad spectrum of possible actions closer to these very target groups. Subject-specific keywords were used to search the literature for vaccine types, drug delivery and formulation in Google Scholar, SciFinder and PubMed. Publications before 2020 were excluded, except for basic research. The PubMed database was used to review the literature from the last 10 years for computer-assisted tools for vaccine development.

## 2. Vaccine Types

Today, there is a broad spectrum of vaccine types to fight well-known diseases and emerging pathogens. This spectrum includes inactivated, live-attenuated, subunit, recombinant, polysaccharide, conjugate toxoid and viral vector vaccines, among others [11]. In view of recent breakthroughs, this review will focus on messenger ribonucleic acid (mRNA) vaccines.

### 2.1. Administration and Targeting Mechanism

Various administration approaches for mRNA lipid nanoparticles (LNPs) have been reported, such as inhalation [12,13], oral [14,15,16], subcutaneous [17], intravenous [18,19,20,21] and intramuscular [22,23,24]. Following administration, the intended targeting mechanism is of great interest. The targeting of mRNA LNPs is beneficial because it can reduce the dose of mRNA required to elicit an immune response and facilitate vaccine production and distribution. To target a specific tissue, the LNP can be modified by directly introducing targeting ligands into the formulation, chemically conjugating them to the LNP surface or by modifying the composition of the lipids in the formulation.

Two different types of targeting are possible. Active targeting can be executed with LNPs containing a target-specific ligand in the formulation, whereas passive targeting can be executed without targeting moiety [25]. Targeting moieties for active targeting mechanisms include antibodies [26], aptamers [27], small molecules [28] and proteins or peptides [29]. Passive targeting (EPR-effect) is influenced by factors such as the size and charge of the LNP, which depend on the molar composition of the different types of lipids used in the formulation [30]. Both mechanisms of active and passive targeting are illustrated in Figure 1.

It is possible to target secondary lymphoid organs by adding phosphatidylserine into the standard four-component MC3-based LNP formulation to facilitate the cellular uptake of immune cells beyond the charge-driven targeting principle commonly used today. As a result, the LNP achieved efficient protein expression in both lymph nodes and the spleen after intravenous administration [32]. Furthermore, reticuloendothelial system (RES)-targeted LNPs were developed by modifying one lipid within the formulation of Onpattro, an approved LNP-based short interfering RNA drug for the treatment of polyneuropathies induced by hereditary transthyretin amyloidosis [33], to switch the surface charge of the LNP from neutral to anionic [34]. Furthermore, it was found that LNPs containing an amide bond in the tail are capable of selectively delivering mRNA to the mouse lung, in contrast to LNPs containing an ester bond in the tail, which tended to deliver mRNA to the liver [35]. Once the cell has been targeted by the LNP, endocytosis is the most investigated internalization mechanism [36].

### 2.2. Endosomale Escape

To induce the efficient release of the mRNA to the cytoplasm after endocytosis, a detailed understanding of the endosomal escape is of major interest. Materials that have the ability to escape from the endosome/lysosome into the cytosol are called endosomolytic agents (e.g., peptides, proteins, toxins, polymers and small chemical compounds) [37]. The modification of LNPs to facilitate endosomal escape is an extensively studied nanotechnology tool for delivering therapeutics into cells. However, the endosomal degradation of LNPs is a major hurdle [38].

Endosomal escape mechanisms include membrane destabilization [39], membrane fusion, proton sponge effect and photochemical internalization. Cationic polymers can escape from endosomes via the proton sponge effect. The downside is that they are covalently linked and usually have low biodegradability and high cytotoxicity [40,41,42]. It was shown that lysosomal cargo escape can be enhanced by excitation, using a lamp or a laser, to induce a more efficient leakage into the cytoplasm. This is the mechanism of photochemical internalization [43].

### 2.3. RNA as the Active Pharmaceutical Ingredient

RNA-Vaccines are currently divided into three main types: non-amplifying mRNA molecules (mRNA), base-modified non-amplifying mRNA molecules (bmRNA), which incorporate chemically modified nucleotides, and self-amplifying mRNA (saRNA or replicons), which maintain self-replicative activity and are derived from an RNA virus vector. saRNA vaccines offer the same benefits as mRNA vaccines, such as fast development, modular design and cell-free synthesis, but require a lower dose due to their self-replicative properties. This allows for quicker production of drug substances and products, which is advantageous during a pandemic response. It also means that a larger percentage of the population can be vaccinated in a shorter period of time [44,45,46].

mRNA is synthesized via in vitro transcription (IVT) with a plasmid DNA (pDNA) template (Figure 2) [47]. An inverted triphosphate cap, such as N7-methylated guanosine, is added to the 5′ end of the mRNA molecule to enhance biological activity [48]. To remove impurities, generated during IVT, the mRNA is purified using chromatography techniques or affinity purification. Finally, a long poly-A tail is added to increase protein expression [49,50].

### 2.4. Role of Adjuvants and LNPs

Adjuvants are useful components by which to enhance the immunogenicity of vaccine formulations. Adjuvants should combine the following biological properties: They should be safe and effective in all age groups. Immune responses should be enhanced in the very young, elderly or immunocompromised populations, where adjuvant activity should be regional and transient, and adjuvants should not directly affect lymphocytes and should not be associated with non-specific B and T cell responses [53]. Non-biological materials containing organic oil, aluminum salt, squalene, liposomes or LNPs are often used to form micro- or nano-sized particles to encapsulate antigens [54]. The formulation of LNPs in mRNA COVID-19 vaccines consists of four main components: (1) a neutral phospholipid, (2) cholesterol, (3) a polyethylene glycol (PEG) lipid and (4) an ionizable cationic lipid. At low pH, ionizable lipids contain positively charged amine groups to encapsulate anionic mRNA during particle formation (Figure 2) and to facilitate membrane fusion during internalization. Particle size can be controlled by the incorporation of PEG-lipids. They also act as a steric barrier to prevent aggregation during storage. Together with the mRNA, these components form particles of about 60–100 nm in size [55]. The size of the LNP, the pKa of the ionizable lipid and lipid gradients affect the tissue and cell specificity of the mRNA vaccine [56]. It is also possible to drive the production of interleukin 6, which leads to a susceptible follicular helper T cell and germinal center B cell response to mRNA and recombinant protein vaccines with ionizable lipid-containing LNPs [57].

### 2.5. Biomaterials Enabled Long-Term Storage

The two main focuses in the formulation development of vaccines are the stability of the final product, which will influence the storage conditions, and the addition of adjuvants to increase their immunogenicity [58]. To enable long-term storage of mRNA vaccines, which can help to increase their availability and accessibility in global health initiatives, the development of new biomaterials and technologies is essential. Cryoprotectants, lyophilization and polymer-based formulations have been determined to increase the self-life of vaccines. Cryoprotectants are substances that protect biological material from damage during freezing and thawing. Trehalose and glycerol are well-known candidates to stabilize mRNA vaccines during long-term storage at low temperatures [59].

LNPs have rapidly gained public attention as a delivery platform for mRNA vaccines. They can be lyophilized and stored for 12 weeks at ambient temperature and for at least 24 weeks at 4 °C without significant changes in physical properties or mRNA delivery efficiency [60]. Compared to LNPs, silica nanoparticles have several advantages that warrant further clinical studies; for example, by modulating their structural and physicochemical properties, such as size, charge, surface functionality and shape, silica nanoparticles can deliver drugs across biological barriers. In addition, silica nanoparticles are stable in harsh biological environments, such as the acidic environment of the stomach, where liposomes typically degrade, limiting their applicability for oral delivery [61]. The lyophilized complex of mesoporous silica nanoparticles with miRNA remained functional after 6 months of storage [62]. Polymer-based formulations, such as hydrogels and microparticles, are used to encapsulate mRNA vaccines and protect them from degradation during long-term storage. These formulations can also be designed to release the mRNA slowly over time, which can improve the immune response and reduce the need for booster shots. The immune response of mRNA LNPs encapsulated in hyaluronan hydrogels was maintained after being stored at room temperature for two weeks [63]. PCL/PLGA/PLLA microspheres were used to deliver SARS-CoV-2 antigens, where no visible particle degradation or changes in porosity patterns were observed during storage at 4 °C for 180 days [64]. For example, a potent self-amplifying RNA (saRNA) vaccine against SARS-CoV-2 that is stable at room temperature has been developed. This saRNA vaccine is formulated with a nanostructured lipid carrier (NLC), which provides stability, ease of manufacture and protection against degradation. Notably, the saRNA/NLC platform demonstrated thermostability when stored lyophilized at room temperature for at least 6 months and at refrigerated temperatures for at least 10 months [65].

## 3. In Silico Tools for Vaccine Development

The three-dimensional structure of a protein determines its properties to a crucial extent. Therefore, techniques to resolve the protein structure are indispensable. However, the experimental determination of a protein structure is very costly and time-consuming, and thus alternative modeling of the structures has gained outstanding importance in basic research. Computer-assisted predictions can be used to model higher-order three-dimensional protein structures based on amino acid sequence data. Acceleration in genome sequencing and the associated identification of protein-coding sequences using diverse bioinformatic methods have increased the available amount of protein data for protein structure prediction.

### 3.1. Epitope Prediction in Immunoinformatics

In pandemic situations, the spread of viruses can be significantly reduced by active immunization, hence, accelerated development based on in silico predictions of envelope protein structures is important, as surface proteins of viruses form the first contact with the immune system. Due to the complexity of the innate and adaptive immune system, many different tools exist to train the immune cells to novel immunogens. The prediction of B-cell and T-cell epitopes has utilized methods in immunoinformatics for many years [66]. The modeling of B-cell epitopes is based on the charged exposed surface area, the secondary structure and on hydrophilicity, as B-cell receptors have primarily hydrophobic binding sites [67]. However, many servers can only identify linear epitopes and not epitopes in which the amino acid residues are in physical contact but are separated in the primary structure [68,69,70]. While the 3D structure of the antigen is used for the identification of these conformational epitopes, the underlying antigen structure must be resolved [71,72]. In contrast, T-cells recognize intracellularly processed antigens in the form of peptides-major histocompatibility complex class (MHC) complexes. The prediction of T-cell epitopes is performed by a variety of methods, including docking models, hidden Markov models, decision trees and artificial neural networks [73,74].

### 3.2. AI-based Protein Complex Predictions

Although machine learning models have been significantly improved in recent years, the prediction of optimal B-cell and T-cell epitopes is often limited due to a low amount of training data [75]. Chen et al. were able to obtain strong T-cell responses by taking advantage of a neural network for predicting antigens in the context of specific human MHC II alleles, called MARIA (major histocompatibility complex analysis with recurrent integrated architecture) [76]. The combination of the mentioned epitope prediction tools and an additional approach, which utilizes good quality data sets to predict 3D protein structures based on its amino acid sequence, would be of significance to adequately target emerging pathogens (Figure 2). Since the accurate determination of the protein structure based on predictions is considered the “holy grail” of protein biochemistry, the Critical Assessment of Structure Prediction (CASP) event, which takes place every 2 years, introduced the AlphaFold neural network in 2020, which appears to master the major challenges of protein structure prediction. AlphaFold combines biological and physical information about protein structures and utilizes multiple sequence alignments to build a deep learning approach, which can predict various protein structures with high accuracy [77]. This neural network developed by DeepMind efficiently learns from the growing experimental data available in the protein data bank, while the AlphaFold prediction significantly accelerates the experimental structure resolution, resulting in an increase in training data [78,79]. In addition to AlphaFold, the neural network RoseTTAFold provides another related approach to solve modeling problems and accurately predict protein complexes [80]. ESMFold and OmegaFold are two more recently published machine learning methods that allow precise predictions of atomic-level protein structures [81,82]. Although AlphaFold was initially limited in predicting multimer complex formation, which is essential for the development of potential vaccine candidates, the published source code was rapidly improved after its release to end up with the prediction tool AlphaFold Multimer [83]. In collaboration with EMBL’s European Bioinformatics Institute, DeepMind made the predictions of over 200 million UniProt entries available, thereby revolutionizing basic and applied research [84]. Whereas the computerized identification of antigens can significantly accelerate vaccine development, the stability of the potentially highly immunogenic protein can complicate expression. Recently, it has been shown that the computational method PROSS can be utilized to increase the stability of the SARS-CoV-2 spike protein while maintaining high immunogenicity [85].

## 4. Digitization and Regulation of Vaccine Quality Management

As described in the previous chapters, in mRNA-based vaccines, the protein-coding sequence is directly introduced into the cells, causing them to express and present the antigen themselves. In contrast to DNA vaccines, recombinant changes in the recipient genome are almost impossible due to the chemical structure of the mRNA sequences [47] and their fast degradation. During the COVID-19 pandemic, more than 90 percent of the 224 million delivered vaccine doses in Germany were based on mRNA systems (by BioNTech/Pfizer 164.6 M doses, Moderna 37.7 M doses) to date [86]. As mRNA contact triggers both humoral and cellular immune response while only entering the cytoplasm, but not the nucleus, mRNA vaccination is seen as highly efficient [87]. The state-of-the-art for the industrial production of the relatively new class of mRNA vaccines is based on enzymatically catalyzed in vitro transcription, a fundamental molecular biology technique. First, plasmids carrying the relevant DNA sequence are produced and linearized. In mRNA manufacturing, the linearized DNA is then enzymatically transcribed into mRNA. After complete processing, the mRNA molecules are encapsulated in LNPs and dosed into storable units during fill-and-finish. Along this manufacturing chain, two quality control steps are implemented regarding DNA quality after linearization and hydrophobic interaction chromatography and mRNA purity after sterile filtration.

### 4.1. Challenges and Benefits of Digital Vaccine Quality Management

Despite its finally recognized medical and economic potential, mRNA vaccine production, like that of most chemical and medicinal products, is still carried out in batches, which limits scalability and associated production capacity. However, the transition to continuous production, as also sought and encouraged by regulatory bodies such as EMA and FDA due to the benefits of agility, quality, flexibility and cost, will only be possible through digital process development and control and digital quality assurance. With digitalization approaches in so-called eQMS (electronic quality management systems; e.g., Veeva Systems, ValGenesis or SAP HANA) on the rise, the trend of digitized research and development, with its increasingly rapid breakthroughs, has also unmistakably impacted the field of pharmaceutical quality control. However, the necessary disruptions in the pharmaceutical industry are lacking, as they already existed in other industries years ago. One challenge could be that established companies (“Big Pharma”) have already invested in existing research, study and manufacturing processes, resulting in a slow drive to invest in new solutions as the risk of low pay-off in time is seen as too high. Pfizer, Sanofi and Bayer helped the respective start-ups BioNTech, Moderna and CureVac to transfer their research on mRNA drugs into COVID-19 vaccines. Conversely, they provided them with the appropriate infrastructure; e.g., pharmacovigilance. In contrast, changing to digital quality management systems from the start of the product life cycle might be able to reduce overall product development costs by up to 20,000 times [88] and promote flexible decision-making towards societal and economic changes, especially underlined by the current pandemic. The pharmaceutical life cycle is composed of the following: product concept, product development, clinical scale manufacturing, commercial manufacturing and market sales. Clinical trials are part of this product lifecycle [89], being the most important to sort out “correlates of immunity/protection” (the number of neutralizing antibodies), whereas study design and cohort recruitment are key to fast approval. In reality, difficulties in recruitment cause delays in 80% of all clinical trials. This might be because not all patients are treated with the drug in the trials (placebo). As recruiting the appropriate patients in sufficient numbers is difficult, they also represent a small portion of the large and diverse population and, accordingly, are not accurate, even though the safety and effectiveness of drugs are crucial aspects evaluated in clinical trials.

### 4.2. Digital Twin Implementations in Vaccine Product Life Cycles

In line with the concept of Quality-by-Design (QbD), the so-called digital twins of a patient could provide different real-time simulations of patients and are thus more comprehensively faster, safer and more accurate. QbD provides the required design spaces to sharpen the quality target product profile (QTPP), related to the quality, safety and efficacy of the active ingredient to avoid out-of-specification batches based on advanced process control using a digital twin. There have only been limited studies made public addressing a digital twin for continuous in vitro transcription that explicitly discuss QbD-compliant model validation and the requirements for a digital twin in the good manufacturing process (GMP)-compliant production [90]. A digital twin is a virtual representation of an object or system that spans its lifecycle, is updated from real-time data and uses simulation, machine learning and reasoning to help decision-making [91]. In this way, the entire production, including quality attributes, process parameters, critical aspects, equipment and processes such as changes, deviations and CAPAs (Corrective Actions, Preventive Actions), can be translated into a virtual image in the digital world (Figure 3).

As in real life, changes and CAPAs are made to this virtual image and deviations occur. The entire life cycle of a product is digitally mapped from the idea generation and conceptualization and a continuous open optimization loop for product and production is introduced, as the entire product life cycle is integrated into factory or plant life cycle and performance data [93]. As an example, periodic reviews of equipment and plants and systems are required by regulation at defined intervals. The evaluation of all changes made during the period under review is considered to assess whether there is an impact on the system. At present, this aspect is “stepmotherly” treated and there is no detailed evaluation of the changes made. In a digital twin, the status could be reviewed and evaluated as to whether the changes had an impact. The operator/system owner is thus supported in the implementation of regulatory requirements. The industry is currently reacting to the momentum of the global digital twin market, estimated at $3.1 billion in 2020 and projected to reach $48.2 billion by 2026, resulting in highly funded endeavors such as the $1.78 million Smart Design and Manufacturing Pilot project between FDA and Siemens to showcase advanced digital design and manufacturing [94].

### 4.3. Regulatory Considerations

Not least, the transition to a digital twin will also have an enormous positive impact on the speed of regulatory processes and approvals. However, accelerated approval was not a new phenomenon of the current COVID pandemic. Laws for accelerated vaccine approval were in power before. Conditional marketing authorization (CMA) in, for example, Germany was introduced with the adaption of regulation (EG) Nr. 726/2004, article 14(7) in 2006 and applied to influenza vaccine approval in 2010 and 2016. The Committee for Medicinal Products for Human Use (CHMP) uses CMA if the benefit-risk balance of the medicine is positive, if it is likely that the applicant will be able to provide comprehensive data post authorization, if the medicine fulfills an unmet medical need or if the benefit of the medicine’s immediate availability to patients is greater than the risk inherent in the fact that additional data are still required. Essentially, the CMPH differentiates into these particular types of procedures in Europe [95]:“Accelerated Assessment”: Medicinal products classified by the CHMP as therapeutic innovations or of major interest to public health and serving an unmet medical need; processing time for marketing authorization application from 210 to 150 days; e.g., Ebola vaccine, BioNTech Pfizer BNT162b2.PRIME (“PRIority MEdicines”): Drugs that serve an unmet medical need or have a clear therapeutic advantage over already approved drugs for the same indication; receive adequate regulatory support to accelerate prior to the official submission of the regulatory dossier; e.g., Ebola and dengue vaccine, Biontech BNT211 cell therapy for the treatment of testicular cancer.“Conditional Approval”: The advantage associated with the market availability of the drug clearly outweighs the potential risks of incomplete data (life-threatening diseases, very rare diseases (“orphan medicines” with lack of data due to low incidence)); e.g., live-attenuated pandemic influenza vaccine H5N1.“Exceptional Circumstances”: Drugs directed against extremely rare target diseases, so that conclusive clinical evidence on safety and efficacy cannot be generated or the collection of the missing data would violate ethical standards; e.g., smallpox vaccine “Imvanex” and inactivated pandemic influenza vaccines.Article 58 procedure: For the scientific evaluation of vaccines not intended for marketing in the European Community to provide expert support for low- to middle-income countries in the regulation and licensing of medicines and to make the high European evaluation standards also available to these countries; e.g., malaria-hepatitis B combination vaccine “Mosquirix”.

On the surface, the relatively swift approval of COVID-19 vaccines under “Accelerated Assessment” with the aid of the centralized “Rolling Review” (RR), where pharmaceutical and non-clinical development data packages are evaluated before the clinical data are fully available, could be dealt as the future of medicinal product approval. However, RR comes with less predictable study outcomes, an overload of regulatory resources and more uncertainties as safety issues may be postponed to the post-marketing phase. Using digital quality assurance, data are recorded without gaps at every point in time, leading to an always up-to-date version of documentation without printing or active distribution necessary and less training effort for the employee. Recording and evaluation could take place in real time [96] and inline continuous documentation of processes leads to clearer comprehension and faster approval through authorities. Our work has shown that few employees in senior positions have deep insight into the regulatory processes of their companies. Most regulatory professionals acting as consultants tend to specialize in the focus of in vitro diagnostics rather than drug product approval, leaving the work of study design and execution to in-house or contracted clinical trial managers. This aspect also affects the regulatory harmonization towards fast approval. Transparency is generally described as an important factor. This involves transparency between stakeholders (strategic data sharing to create a trusting partnership) [97], transparency of supply chains (as the potential for improvement and to enable better forecasting and risk management) [98,99] and socially appropriate knowledge communication to prevent vaccination hesitancy. Undeniably, there is potential for greater transparency in government-manufacturer agreements, which would be beneficial to both planning for suppliers and to increasing public trust [100].

## 5. Discussion

In this review, we identified the bottlenecks of the conventional vaccine production chain to map them against current innovation approaches (Figure 4). To do this, we first conducted an ecosystem and stakeholder analysis and laid out the losses and gains of the key players in a value proposition canvas. As a result, we derived a concept that focuses on four main drivers to accelerate the vaccine product life cycle: computerized candidate screening, modular production, digitized quality management and a resilient business model with corresponding transparent supply chains.

First, we highlighted the advantages and disadvantages of mRNA-LNP-vaccines; it is important to point out that there is no “panacea” and it will be essential to rely on a variety of vaccine-types to encounter future pathogens.

The need for rapid adaptation to a rapidly changing environment makes mRNA-LNP vaccines a promising tool due to their modular production capabilities in terms of formulation and drug delivery. It is possible to tailor the individual components such as the mRNA for the expression of identified antigens (or proteins in general) to elicit an immune response and the lipid components to target specific tissues by passive or even active targeting mechanisms to minimize off-target effects and encapsulate the mRNA cargo for prolonged circulation in the bloodstream.

In addition, it is possible to produce mRNA-LNP vaccines on a large scale by simply mixing the required components for the rapid distribution of sufficient vaccines for global use. As the development of vaccines that are stable at elevated temperatures continues, production time and costs will decrease, and distribution routes will be simplified.

The incorporation of improved micro-mixing technology and in-line DLS (Dynamic Light Scattering) into the modular production platform enables the production of precisely tailored LNPs in terms of particle size and particle size distribution, with the possibility of upscaling. Furthermore, this technology can be used to produce small quantities of drug products for efficient vaccine candidate screening. Further research should focus on understanding the mechanisms of internalization. This may not only help to improve drug delivery and targeting, as well as to reduce the required drug dose to minimize cost and production time, but may also reduce off-target effects.

During the Corona pandemic, effective vaccines against the SARS-Cov-2 virus could be developed in a relatively short time. A key factor was the knowledge of the resolved protein structure of the SARS-CoV virus, which could be transferred to SARS-CoV-2 [101]. For this purpose, a 2P modification was used, which utilizes two proline mutations to stabilize the pre-fusion conformation, resulting in an efficient neutralizing antibody response [102]. Because habitats of currently separated species will overlap in the future and pathogens can consequently infect new hosts, pandemic situations, and thus novel viruses, will play a major role. In addition, the rapid evolution of viruses and their adaptation to selection pressures is a particular challenge to vaccine development, so the accelerated development and adaptation of vaccines are urgently needed. Therefore, the recently developed approaches for accurate protein structure prediction can make a substantial contribution by rapidly responding to future pandemics by modeling potential vaccine candidates.

In terms of regulatory aspects and quality management, various emergency tools for accelerated vaccine approval were already given before the current pandemic, but are in constant transit in view of the relatively new mRNA technology and emerging pathogens. “Accelerated Assessment” under Rolling Review makes sense as an instrument in emergencies where benefits outweigh risks, but it remains questionable as a standard instrument. This would mean that far more human capacity (headcounts) would have to be mobilized in order not to permanently overburden resources on the part of the authorities. Systemic and digitalized solutions, i.e., digital twin technology both in companies and authorities, should be suggested, but only after research has led to reliable functionality for human use. Particularly, in the ethically highly sensitive area of vaccines, in accordance with the principle of medical practice “primum non nocere”, no compromises must be made in streamlining regulatory processes. It is worth emphasizing that active immunization and studies of (new) vaccine products always involve healthy individuals, which raises an ethical sticking point. For this reason, psychological research fields in vaccine hesitancy have also been very active [103].

The EMA aims towards regulatory research and innovation drive in their “EMA Regulatory Science to 2025—Strategic reflection” and the International Council for Harmonization of Technical Requirements for Pharmaceuticals for Human Use (ICH) promotes the digital and fast exchange of approval documents to be accepted among the authorities that take part in the ICH. In the case of the BioNTech COVID vaccine, the EMA gave data to the FDA. RR could also be supported by digital solutions/automated document uploads, and headcounts would only be necessary for decision-making. In conclusion, the digital twin integration of equipment and process is most urgent today for increasing the productivity and profitability of enterprises and the quality of products. Today, many companies are engaged with the methodological issues of building digital twins, determining a set of controlled parameters/signals and their classification. Further research in the field of digital twins should be aimed at organizing communication between a real industrial facility and its digital model; i.e., the exchange of data between real and virtual controllers. Thus, the modern approach to quality management is increasingly becoming a digital version of the quality management system, which is driven by artificial intelligence. To adapt to the short product life cycle, but not overburden employees beyond their capacity, we need to move away from human-driven production and support the trend of digitalized process control and QA. Clearly, the enormous number of researchers attracted to the COVID-19 field of work may also have contributed to the rapid advancement of vaccines [104,105], and now the same attraction is needed at the level of pharmaceutical digitization.

## 6. Conclusions

In reality, vaccine development and adaptation, especially in the area of mRNA technology, is not necessarily a truly circular or linear product life cycle, as suspected at the beginning. Rather, our investigations showed that the communication interfaces between the individual instances within the process fields require major improvements. Comparable to other industries, the concatenation of already existing innovations and the associated digital knowledge transfer between basic research and application is the key to rapid, demand-driven vaccine production. In the future, this review is to give thought and aid to combining these multifaceted technological breakthroughs.

## Figures and Tables

**Figure 1 vaccines-11-01287-f001:**
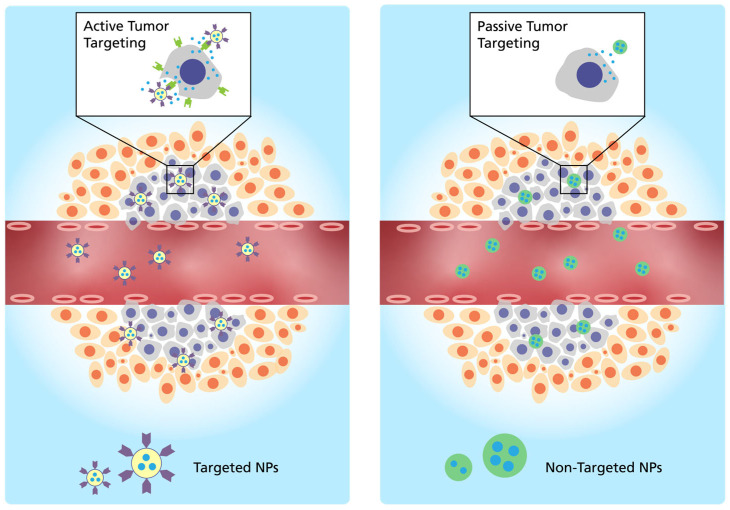
Targeting mechanisms of mRNA-LNPs: (**left**) Active tumor targeting executed with a target-specific ligand in the formulation (antibodies, aptamers, small molecules and proteins or peptides), (**right**) passive tumor targeting based on the EPR-effect (adapted from [31]).

**Figure 2 vaccines-11-01287-f002:**
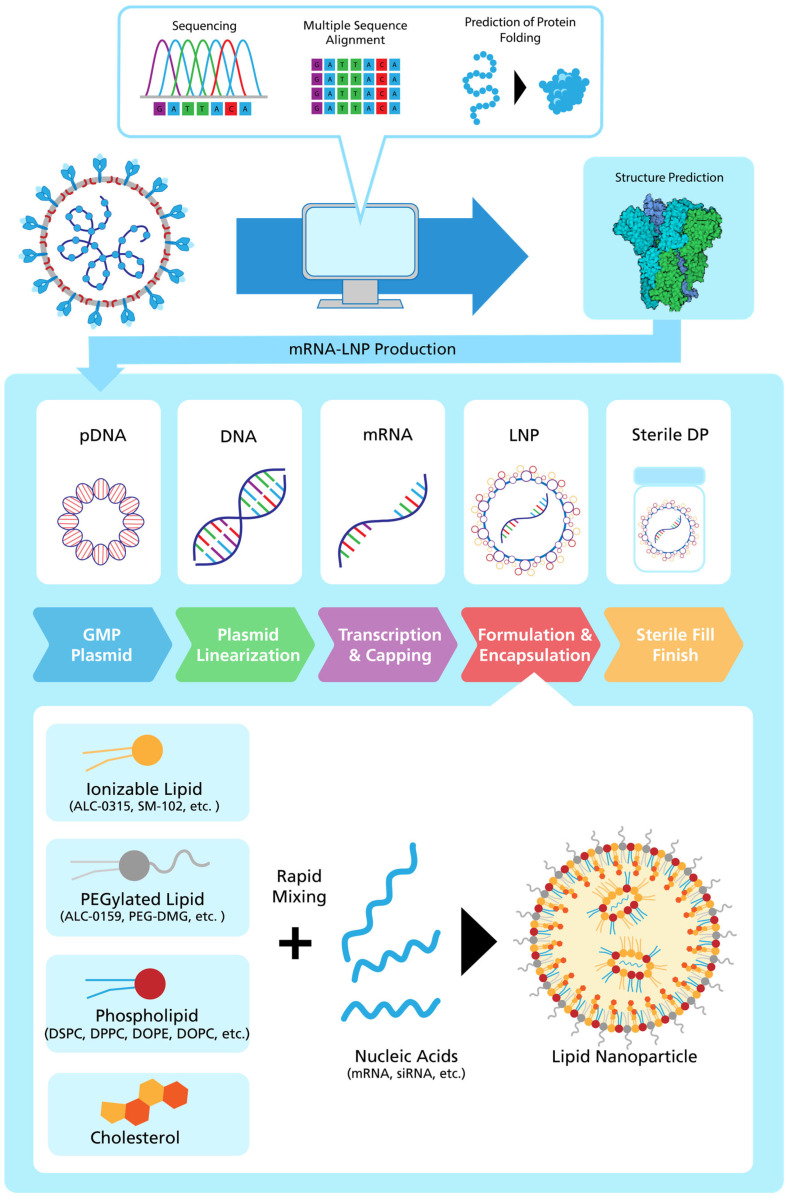
Process for mRNA-LNP vaccine development: (**top**) Computer-assisted prediction of coronavirus spike protein (PDB: 5I08), (**middle**) mRNA-LNP production [50] starting from the manufacturing of plasmid DNA (pDNA), mRNA and LNP to the sterile drug product (DP) (adapted from [51]), (**bottom**) Lipid composition of mRNA-LNPs (adapted from [52]).

**Figure 3 vaccines-11-01287-f003:**
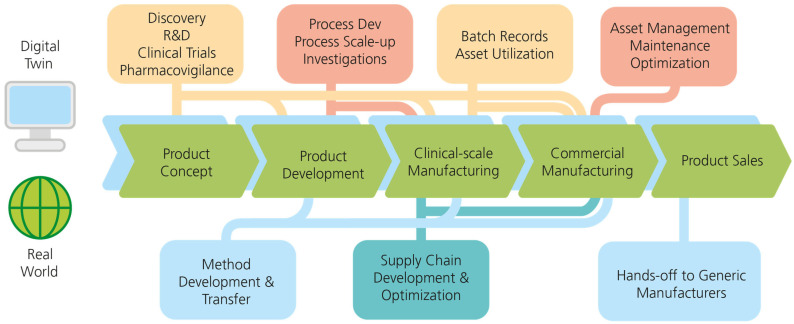
Real and digital production. A digital twin offers actionable process steps at all stages of the production process chain compared to the real-world scenario (modified from [92]).

**Figure 4 vaccines-11-01287-f004:**
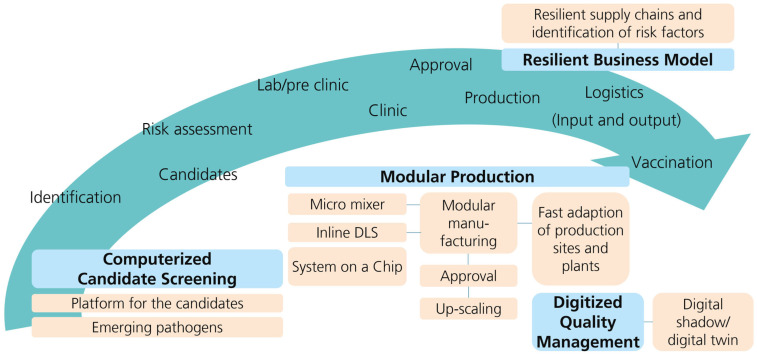
Main drivers to accelerate the vaccine product life cycle: computerized candidate screening, modular production, digitized quality management and a resilient business model with corresponding transparent supply chains.

## Data Availability

No new data were created or analyzed in this study. Data sharing is not applicable to this article.

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
