# Peer review of "Just Keep Rolling?—An Encompassing Review towards Accelerated Vaccine Product Life Cycles"

_vaccines, 2023, doi:10.3390/vaccines11081287_

Round 1
Reviewer 1 Report
The manuscript presents a grouping of methodological approaches that can be employed to accelerate and scale up vaccine production in urgent situations such as a pandemic. Although the review deals superficially with numerous biological strategies and definitions, the discussion corroborates the rationale. The manuscript does not exhaust the subject, and therefore we feel the need for some corrections/updates of the issue and literature are necessary.
1) Item 3.1-Line 224-240. “Epitope prediction”- We agree that immunoinformatics has been used for years to predict linear B and T epitopes. Although new algorithms have been developed in the last few decades regarding identifying linear B epitopes, these methodologies still fall short. They usually identify (1) long sequences, not accurately informing the beginning and end of the epitope, (2) they do not inform which immunoglobulin class binds to the epitopes, and this is important concerning the identification of blocking and protective antibodies, and (3) have a very low degree of accuracy, only 40-50% for linear B epitopes. Regarding identifying T epitopes, the degree of success is already much higher.
Therefore, there are peptide microarray methodologies whose identification success is greater than 98%, and the time to obtain these experimental results is much faster than that of immunoinformatics, so we suggest that this paragraph be completely updated. There are numerous examples in the literature in this regard.
2) Line 280- the reference was missing.
Author Response
We would like to thank the reviewer for the helpful feedback and have made changes accordingly. Through the helpful review reports, we believe we have improved the quality of our review such that it is now ready for publication. All the modifications made in manuscript are marked in yellow:
1) Item 3.1-Line 224-240. “Epitope prediction”
- Thank you for pointing this out. We agree that peptide microarray methodologies have a significant impact on accelerating vaccine development. However, the focus of this section is on "In silico tools for vaccine development". In addition, there are other experimental approaches that are important for vaccine development but are beyond the scope of this review. Therefore, as far as possible, this section will focus only on computer-based methods.
2) Line 280- the reference was missing.
- Thank you. The reference is: Federal Ministry of Health. Current vaccination status. Available online: https://impfdashboard.de/en/ (accessed on 17 May 2023).
Reviewer 2 Report
This review aims to emphasize recent approaches for targeted rapid adaptation and production of vaccines from an interdisciplinary multifaceted perspective. A few concerns should be addressed:
(1) The information about the search strategies, data sources (Web of Science, Scopus, or other databases; Reasons for the choice of data source), and data coverage years should be provided.
(2) The sheer number of researchers flocking to Covid-related research may also have accelerated the search for vaccines. The authors can refer to the following two studies.
(a) Wagner, C. S., Cai, X., Zhang, Y., & Fry, C. V. (2022). One-year in: COVID-19 research at the international level in CORD-19 data. PLoS One, 17(5), e0261624.
(b) Liu, W., Huangfu, X., & Wang, H. (2023). Citation advantage of COVID-19-related publications. Journal of Information Science, 01655515231174385.
(3) Please correct “Error! Reference source not found”
(4) References 40 and 41 are identical
OK
Author Response
We would like to thank the author for the helpful feedback and have addressed the concerns accordingly. Through the helpful review reports, we believe we have improved the quality of our review such that it is now ready for publication. All the modifications made in manuscript are marked in yellow:
(1)
- As you suggested we add the information of the search strategies in the manuscript. Subject specific keywords were used to search literature for vaccine types, drug delivery and formulation containing: “(formulation of) mRNA-vaccines, (transfection, endosomal escape of) lipid nanoparticles, (targeted, active, passive) drug delivery (vehicles), adjuvants, vaccine storage and immunogenicity” in google scholar, sci-finder and PubMed. Publications before 2020 were excluded, except for basic research. The PubMed database was used to review literature from the last 10 years for computer-assisted tools for vaccine development. The PubMed database was used to review literature from the last 10 years for computer-assisted tools for vaccine development. Therefore, literature searches based on title, abstract, and keywords utilizing "vaccine” and ”predict" as search terms were executed.
(2)
- Thank you for your suggestion. We added the REFs in the discussion accordingly:
“Clearly, the enormous number of researchers attracted to the COVID-19 field of work may also have contributed to the rapid advancement of vaccines [97,98] and now the same attraction is needed at the level of pharmaceutical digitization.”
(3) (4)
- Thank you for pointing out the issue, we adapted the REFs.
Reviewer 3 Report
very interesting article - though largely limited to mRNA vaccines - may have a good impact on other vaccine productions
Some references are not available - that has to be corrected
Author Response
We would like to thank the referee for the positive feedback. Through the helpful review reports, we believe we have improved the quality of our review such that it is now ready for publication.
Some references are not available - that has to be corrected
- Thank you for pointing out the lacking information that we added accordingly. All the modifications made in manuscript are marked in yellow.
Reviewer 4 Report
It is a very interesting paper. It is well structured and very well presented with the accompanying figures. The authors explain the procedure for the development and manufacture of vaccines taking into account the particular circumstances of the Covid19 pandemic.
I only have some considerations to make, which, in my opinion, can improve the focus of the work.
Sections 2 and 3 of the work I see excessively developed. However, I believe that section 4 should be expanded as a novelty in this work and, specifically, section 4.3 on regulatory considerations should be expanded to include european regulations.
In the discussion section on page 11 of 19, the paragraph included between lines 421 and 429 should be explained given its importance for the future.
Some comments on the ethical aspects of regulatory research and driving innovation should also be included in this section.
Author Response
We would like to thank the referee for the positive feedback. Through the helpful review reports, we believe we have improved the quality of our review such that it is now ready for publication. All the modifications made in manuscript are marked in yellow.
Sections 2 and 3 of the work I see excessively developed. However, I believe that section 4 should be expanded as a novelty in this work and, specifically, section 4.3 on regulatory considerations should be expanded to include european regulations.
- Thank you for highlighting this room for improvement. We inserted relevant information including European regulations on p. 10 f:
“The Committee for Medicinal Products for Human Use (CHMP) uses CMA if the benefit-risk balance of the medicine is positive, if it is likely that the applicant will be able to provide comprehensive data post-authorization, if the medicine fulfils an unmet medical need or if the benefit of the medicine's immediate availability to patients is greater than the risk inherent in the fact that additional data are still required. Basically, the CMPH differentiates into these particular types of procedures in Europe [92]:
- "Accelerated Assessment": Medicinal products classified by the CHMP as therapeutic innovations or of major interest to public health and serving an unmet medical need; processing time for marketing authorization application from 210 to 150 days; e.g. Ebola vaccine, BioNTech Pfizer BNT162b2.
- PRIME ("PRIority MEdicines"): Drugs that serve an unmet medical need or have a clear therapeutic advantage over already approved drugs for the same indication; receive adequate regulatory support to accelerate prior to official submission of regulatory dossier; e.g., Ebola and dengue vaccine, Biontech BNT211 cell therapy for treatment of testicular cancer.
- "Conditional Approval": The advantage associated with the market availability of the drug clearly outweighs the potential risks of incomplete data (life-threatening diseases, very rare diseases ("orphan medicines" with lack of data due to low incidence)); e.g. live-attenuated pandemic influenza vaccine H5N1.
- "Exceptional Circumstances": Drugs directed against extremely rare target diseases, so that conclusive clinical evidence on safety and efficacy cannot be generated or the collection of the missing data would violate ethical standards; e.g. smallpox vaccine "Imvanex" and inactivated pandemic influenza vaccines.
- Article 58 procedure: For the scientific evaluation of vaccines not intended for marketing in the European Community to provide expert support to low to middle income countries in the regulation and licensing of medicines and to make the high European evaluation standards also available to these countries; e.g. malaria-hepatitis B combination vaccine "Mosquirix".
On the surface, the relatively swift approval of COVID-19 vaccines under „Accelerated Assessment“ with the aid of the centralized “Rolling Review” (RR), where pharmaceutical and non-clinical development data packages are evaluated before the clinical data is fully available, could be dealt as the future of medicinal product approval. However, RR comes with less predictable study outcome, an overload of regulatory resources and more uncertainties as safety issues may be postponed to the post-marketing phase.”
In the discussion section on page 11 of 19, the paragraph included between lines 421 and 429 should be explained given its importance for the future.
Some comments on the ethical aspects of regulatory research and driving innovation should also be included in this section.
- Thank you for pointing at the possible improvements concerning regulatory. Indeed, regulatory research and adaption was markedly pushed during the Covid19 pandemic. We have added additional sentences to this topic. However, we also believe that regulatory tools were partly installed before the pandemic and more so improvements in this area represent a consequence of the exploding biomedicinal breakthroughs, rather than a novelty. What is important is the digitalization disruption and the digitalization research transfer from industrial areas (engines, plants) on the pharmaceutical sector.
“Our work has shown that few employees in senior positions have deep insight into the regulatory processes of their companies. Most regulatory professionals acting as consult-ants tend to specialize in the focus of in vitro diagnostics rather than drug product approval, leaving the work of study design and execution to in-house or contracted clinical trial managers. This aspect also affects the regulatory harmonization towards a fast approval. Transparency is generally described as an important factor. This involves transparency between stakeholders (strategic data sharing to create a trusting partnership) [REF], transparency of supply chains (as potential for improvement and to enable better fore-casting and risk management) [REF,REF], and socially appropriate knowledge communication to prevent vaccination hesitancy. Undeniably, there is potential for greater transparency in government-manufacturer agreements, which would be beneficial both to planning for suppliers and to increasing public trust [REF].”
“Systemic and digitalized solutions i.e. digital twin technology both in companies and authorities should be suggested, but only after their research has led to reliable functionality for human use. Particularly in the ethically highly sensitive area of vaccines, in accordance with the principle of medical practice "primum non nocere", no compromises must be made in streamlining regulatory processes. It is worth emphasizing that active immunization and studies of (new) vaccine products always involve healthy individuals, which raises an ethical sticking point. For this reason, psychological research fields on vaccine hesitancy have also been very active [96].”
Round 2
Reviewer 1 Report
The indicated suggestions were fulfilled and now we consider that the paper can be accepted for publication,
Reviewer 2 Report
OK